# Clinical and Epidemiological Characteristics of Severe Acute Adult Poisoning Cases in Martinique: Implicated Toxic Exposures and Their Outcomes

**DOI:** 10.3390/toxics8020028

**Published:** 2020-04-09

**Authors:** Dabor Resiere, Hatem Kallel, Odile Oxybel, Cyrille Chabartier, Jonathan Florentin, Yannick Brouste, Papa Gueye, Bruno Megarbane, Hossein Mehdaoui

**Affiliations:** 1Intensive Care Unit, University Hospital of Martinique, Fort-de-France, 97261 Martinique, France; odile.oxybel@chu-martinique.fr (O.O.); cyrille.chabartier@chu-martinique.fr (C.C.); hossein.mehdaoui@chu-martinique.fr (H.M.); 2Intensive Care Unit, Cayenne General Hospital; 97300 Cayenne, French Guiana, Kallelhat@yahoo.fr; 3Department of Emergency Medicine, University Hospital of Martinique, Fort-de-France, 97261 Martinique, France; jonathan.florentin@chu-martinique.fr (J.F.); yannick.brouste@chu-martinique.fr (Y.B.); 4Emergency Medical Services (Service d’aide médicale d’urgence 972), 97261 Martinique, France; papa.gueye@chu-martinique.fr; 5Department of Medical and Toxicological Critical Care, Lariboisière Hospital, Paris-Diderot University, INSERM UMR-S 1144, 75013 Paris, France; bruno.megarbane@aphp.fr

**Keywords:** epidemiology, acute poisonings, intensive care unit, Martinique

## Abstract

The epidemiology of severe acute poisonings in the French overseas departments of the Americas remains poorly reported. The main objective of this study was to determine the epidemiology and characteristics of severe acutely poisoned adult patients. Methods: A retrospective descriptive study was conducted from 1 January 2000 to 31 December 2010 in severely poisoned patients presenting to the emergency department (ED) of the University Hospital of Martinique, and the general public hospitals of Lamentin and Trinité. Results: During the study period, 291 patients were admitted for severe poisoning, giving an incidence rate of 7.7 severe cases/100,000 inhabitants. The mean age was 46 ± 19 years and 166 (57%) were male. Psychiatric disorders were recorded in 143 (49.8%) patients. Simplified Acute Psychological Score (SAPS II) at admission was 39 ± 23 points and Poisoning Severity Score (PSS) was 2.7 ± 0.8 points. Death was recorded in 30 (10.3%) patients and hospital length of stay was 6 ± 7 days. The mode of intoxication was intentional self-poisoning in 87% of cases and drug overdose was recorded in 13% of cases. The toxic agent involved was a therapeutic drug in 58% and a chemical product in 52% of cases. The predominant clinical manifestations were respiratory failure (59%), hemodynamic failure (27%), neurologic failure (45%), gastrointestinal manifestations (27%), and renal failure (11%). Polypnea, shock, ventricular fibrillation or tachycardia, and gastro-intestinal disorders were the main symptoms associated with death. The main biological abnormalities associated with death in our patients were metabolic acidosis, hypokalemia, hyperlactatemia, hypocalcemia, renal injury, rhabdomyolysis, increased aspartate aminotransferases, and thrombocytopenia. Extracorporal membrane oxygenation (ECMO) was used in three patients and specific antidotes were used in 21% of patients. Conclusions: Acute poisonings remain a major public health problem in Martinique with different epidemiological characteristics to those in mainland France, with a high incidence of poisoning by rural and household toxins.

## 1. Introduction

Acute poisonings are adverse health effects due to acute exposure (less than 24 hours) to a toxic substance, medications, drug overdose, acute drug abuse problems, chemical exposure, occupational and environmental toxins, biological agents, and envenomations [1,2]. Intoxications are major public and preventable health issues, which are a frequent reason for the consultation of emergency rooms [3,4] and typically represent 1–3% of all emergency department (ED) visits. Severe forms needing admission to intensive care units (ICUs) account for four and up to 40% of cases [5,6,7] with a mortality rate of 3 to 6% [2,8,9,10,11,12].

Therapeutic drug intoxication and, in many countries, chemical products such as pesticides remain a common cause for consultation in the ED and a significant cause for ICU admissions and fatalities. Indeed, severe cases with life-threatening conditions needing technical support or requiring constant monitoring are usually hospitalized in the ICU. Furthermore, the management of acute poisonings includes examining the mental health and social issues that lead to severe poisoning to help reduce the recurrence and poisoning fatality rates.

The toxic agents and the pattern of acute poisoning vary between centers in various countries that can change over time. Therefore, there is a constant need to obtain up-to-date information on acute poisoning for planning the rational use of resources and evaluating public health interventions. In most developed countries, anti-poison centers are available 24 h to record information and to provide specific and appropriate toxicological advice to manage poisoned patients. The epidemiologic data obtained permits a survey of the main trends of poisonings and the toxins used and to implement preventive actions. Thus, in metropolitan France, the epidemiology of acute poisonings is subject to frequent updates from the registries of the anti-poisons and toxicovigilance centers. For the first time, in 2006, 197,042 cases of human exposure have been analyzed and published [13]. Interestingly, the vast majority (82.5%) of poison exposure was unintentional. In Martinique, however, a comparable database is sorely lacking. Few scientific data are therefore available about the characteristics of acute poisonings in the Caribbean region.

The aim of this study was to describe the clinical and sociodemographic patterns of acute poisonings in the EDs of the three main hospitals of Martinique (University Hospital of Martinique, and the general hospitals of Trinité and Lamentin).

## 2. Methods

Our study was retrospective and multicentric. It was conducted over 11 years from January 2000 to December 2010 in the three main hospitals of Martinique (University Hospital of Martinique and the general hospitals of Trinité and Lamentin), according to the Helsinki principles and it was declared to our national database review board (N°: 2216995 v 0).

Martinique is an overseas, insular region of France located in the Lesser Antilles of the West Indies in the eastern Caribbean Sea. One of the Windward Islands, it is directly north of Saint Lucia, northwest of Barbados and south of Dominica. The land area is about 1128 square kilometers and the population size is 376,480 inhabitants as of January 2016.

We included all patients aged more than eighteen years old presenting with severe poisoning. We excluded all patients less than eighteen, those presenting following envenomation by animal bites and stings, and those whose records were missing or did not provide sufficient data for the analysis. All patients admitted to the University Hospital of Martinique, the general hospital of Trinité, or the general hospital of Lamentin, whose primary or secondary diagnosis was acute poisoning, were initially selected from the archives of the Department of Medical Information (DIM) of these three hospitals. The selection of charts was based on the 10^th^ revision of the International Statistical Classification of Diseases and Related Health Problems (ICD-10). The codes used were poisoning by adverse effect of and underdosing of drugs, medicaments, and biological substances (T36-T50) and toxic effects of substances chiefly nonmedicinal as to source (T51–T65). Only charts with severe acute poisoning were studied. We recorded epidemiological and clinical data including age and sex of the patients, the date and time of the poisoning, the toxic substance, the type of exposure, and the clinical and biological parameters at admission to the ED.

Acute poisoning is defined as adverse health effects due to acute exposure (less than 24 hours) to a toxic substance, medications, drug overdose, acute drug abuse problems, chemical exposures, occupational and environmental toxins, biological agents, and envenomings [1,2]. Severe acute intoxication is defined as an acute poisoning requiring technical support or close surveillance [1].

Toxic agents were classified as therapeutic drugs and chemical drugs. The subgroup of toxic agents was categorized based on the indications for use. Multiple drugs were defined as the ingestion of two or more drugs. The type of exposure was classified as intentional self-poisoning, overdosage, accidental, or unknown [14]. Severity at admission to ICU was assessed using the Simplified Acute Physiology Score (SAPS II) [15] and the severity of the poisoning was assessed using the Poisoning Severity Score (PSS) [16].

## 3. Statistical Analysis

Results were reported as mean ± standard deviation or numbers with percentages. Bivariate statistical comparisons were conducted using the Chi-square or Fisher’s exact test for categorical data and the Independent-Samples T-test for continuous data. Non-redundant variables selected by bivariate analysis (*p* < 0.05) and considered clinically relevant were entered into a logistic regression model. Results were expressed as odds ratios (OR) with their 95% confidence intervals (CI). A *p*-value < 0.05 was considered statistically significant. 

Receiver operating characteristic (ROC) curves were used to evaluate the diagnostic value of quantitative tests. The area under the curve was estimated by the method of Hanley and McNeill [17]. All statistical analyses were carried out with Excel (2010 Microsoft corporation, Redmond, USA) and IBM SPSS Statistics for Windows, version 24 (IBM Corp., Armonk, NY, USA, 2018).

## 4. Results

During the study period, 291 patients were admitted for severe poisoning, giving an incidence rate of 7.7 severe cases/100,000 inhabitants. The mean age was 46 ± 19 years and 166 (57%) were males. A history of psychiatric disorders was recorded in 143 (49.8%) patients. SAPS II at admission was 39 ± 23 points and PSS was 3 ± 1 points. Thirty patients (10.3%) died, and hospital length of stay was 6 ± 7 days. Table 1 reports the epidemiological characteristics and outcomes of these patients.

The mode of intoxication was intentional self-poisoning in 250 (86.5%) cases. The toxic agent involved was a therapeutic drug in 169 (58.1%) cases and a chemical product in 152 (52.2%) cases. A multiple therapeutic drugs intoxication was recorded in 61 (36.1%) cases, a multiple chemical intoxication was recorded in 48 (31.6%) cases, and a mixed therapeutic and chemical toxic was recorded in 30 cases (10.3%). Drug overdose was recorded in 37 (12.7%) cases. The most involved therapeutic drugs were psychotropic drugs (116 cases), benzodiazepines (76 cases), neuroleptics (33 cases), cardiotrops (26 cases), and tricyclic antidepressants (18 cases). The most involved chemical toxics were OH3 (56 cases), herbicide (29 cases), Rubigine^®^ (28 cases), caustics and Aldicarb (23 cases each) (Figure 1). Table 2 compares the survivors and non-survivors in relation to the various toxic substances used.

Clinical manifestations dominated by respiratory failure in 171 (58.8%) patients, hemodynamic failure in 78 (26.8%) patients, neurologic failure in 130 (44.7%) patients, gastrointestinal manifestations in 78 (26.8%) patients, and renal failure in 32 (11%) patients. The mean age was 56 ± 18 years in non survivors and 45 ± 18 years in survivors (*p* = 0.001). Figure 2 shows the mortality rate according to age range and Figure 3 shows the accuracy of age to predict death in poisoned patients. Polypnea, shock, ventricular fibrillation or tachycardia, and gastro-intestinal disorders are the main symptoms associated with death. The main biological abnormalities associated with death in our patients were metabolic acidosis, hypokalemia, hyperlactatemia, hypocalcemia, renal injury, rhabdomyolysis, increased aspartate aminotransferases, and thrombocytopenia. Table 3 reports the clinical and biological parameters recorded in our patients. SAPS II score and PSS were strongly predictive of severe outcome (Figure 4).

Therapeutic management included symptomatic measures as well as antidotes and life-support techniques. Mechanical ventilation was needed in 171 (58.8%) patients, catecholamines in 78 (26.8%) patients, renal replacement therapy in 32 (11%), and ECMO in three patients. Specific antidotes were used in 60 (20.6%) patients. Table 4 shows the therapeutics used in our patients. 

## 5. Discussion

Our study provides information on the clinical and epidemiological characteristics of patients with severe acute poisoning in Martinique. It shows that the incidence rate of severe poisoning is high in this French overseas territory. Rates of intentional self-poisoning in Martinique were highest in males than in females. The common toxic agents recorded were chemical drugs, but also specific toxins that are forbidden according to European and French Law.

Acute poisonings are epidemic worldwide. They are a frequent reason for the consultation of emergency rooms [3,4], represent 1–3% of all EDs visits, and account for 10% of all admissions for injuries [18]. Annual hospital admissions for poisonings are about 260,000 in the USA and 80,000 in England [19,20]. Overdoses account for one-fourth of all suicide attempts [20]. In the USA, the rate of deaths from drug overdoses has increased from 4.4 per 100,000 population in 1999 to 14.7 in 2014 involving opioids [21]. Admissions to ICU for drug overdoses vary from country to country and account for approximately 2.5–14% of poisoned patients [10,12,22]. Our study including only severe patients showed an incidence rate of 7.7 severe cases/100,000 population. This public health problem can also be responsible for high mortality (10%) and disability in young people.

Victims of acute poisoning are typically young women with a history of psychiatric disorders [3,23]. In our study, the mean age was 46 ± 19 years and 57% of patients were male. A history of psychiatric disorders was recorded in 50% of our patients. Our results showed that patients at risk in Martinique are young people with a predominance of males and people with previous psychiatric disorders. The majority of deaths were related to attempted suicide (86.5%) and the mean age of non-survivor patients was 56 ± 18 years. These data are slightly different from those of metropolitan France, where 65% of fatal poisonings are self-inflicted and the age group most represented is that of 30 to 59 years [1]. The severity of cases in our study is evidenced by respiratory failure (59%), hemodynamic failure (27%), neurologic failure (45%), and renal failure (11%).

The prognosis of severely poisoned patients is generally favorable with a typically short ICU and hospital stay. Indeed, mortality rates in the ICU for poisoning admissions range from 1 to 6%, according to most reports [10,11,12,22,24,25] and are significantly lower than that of the general ICU population [26,27,28,29,30,31,32]. About 60% of the patients can leave the ICU within 24 h and only approximately 11% require critical care for longer than 48 h [7,33,34]. In a Dutch study [25], intoxications by opiates, cocaine, and amphetamine had the highest mortality two years after ICU admission (12.3%), whereas a combination of different intoxications had the lowest (6.3%). Additionally, mortality increases with age [25,35].

In our study, mortality rate was 10.3%, the highest reported in the literature for other settings [10,11,12,22,24,25]. Factors associated with severe outcomes were organ failure, metabolic acidosis, hypokalemia, hyperlactatemia, hypocalcemia, rhabdomyolysis, increased aspartate aminotransferases, and thrombocytopenia. These factors are commonly associated with death in critically ill patients in general. Age was higher in survivors. However, the Receiver operating characteristic curve analysis showed a low value of age to predict death in severely poisoned patients (AUC: 0.678), but the mortality rate was higher in patients greater than eighty years old.

Intoxications with ethanol or drugs are the most frequent causes in industrialized countries [23,36,37,38,39]. In a Swedish study including 8155 intoxicated intensive care patients, the leading causes of admission were “mixed intoxications” (30%), intoxication to sedatives, hypnotics, and anticonvulsants (16%) and ethanol intoxication (15%) [39]. In a French observational study including about 2500 patients, antidepressants, neuroleptics, benzodiazepines, and heart medication were the leading causes of intoxication-related ICU admission [23]. However, in developing countries, intoxications with pesticides and organophosphates are important causes leading to ICU admission [34,40]. 

In our study, the majority of serious poisonings and deaths were related to psychotropic and cardiotropic drugs (Table 5). These data are comparable to those observed in metropolitan France and other industrialized countries [23,36,37,38,39]; however, they can also be related to other toxic agents like pesticides. Indeed, in Martinique, severe toxicity and subsequent death can result from exposure to specific toxicants such as non-authorized drugs including industrial chemicals and herbicides (paraquat) or some local fruits and plants [41,42]. Deaths secondary to pesticide poisoning represent 22%, comparable to the epidemiology reported from developing countries (23% of self-inflicted poisoning in Africa and 21% in South-East Asia) [20]. Another herbicide causing severe intoxications and death in Martinique is Rubigine^®^, which accounted for 16% of all deaths in our study (slightly more than one-quarter of deaths due to non-medicinal drugs). Martinique consumes large quantities of pesticides (6 kg per capita of pesticides imported in 2002), mostly used in banana plantations. The use and storage of these pesticides by professionals, but also illegally by individuals, can lead to the occurrence of acute poisoning.

In our study, there was no serious accidental poisoning by pesticides or herbicides. Nor were there occupational poisonings. It may be concluded that the precautions or prohibitions relating to the use of these products are sufficiently respected in the agricultural setting in Martinique. Pesticide poisonings found in our study were mostly domestic. This implies that individuals still have access to these dangerous products, ordinarily accessible only to agricultural professionals, according to the decree of 26 February 2004 in France. Two products were most commonly found in our series: Aldicarb and Paraquat. Aldicarb is a carbamate insecticide that is the active substance in the pesticide Temik^®^. It was the most commonly used pesticide for severe poisoning in our study. It was prohibited by Prefectural decree in Martinique in 2002 and withdrawn from the market in 2004 at the national level. However, the product is still in circulation on the island either due to non-elimination and non-collection of existing stocks from agricultural professionals or because of the illegal acquisition of the product. This product is banned in France and the European Union, but is still available in certain regions of the world, particularly in the United States (where the final and total withdrawal of the product was scheduled for 2018) and in the Caribbean islands (the Dominican Republic and Saint Lucia in particular). In Martinique, it is used by the population as a rodenticide. Despite the frequency of severe poisoning by Temik^®^, there were no related deaths in our study.

In our study, only one patient presented with crack, and four with cocaine poisoning. However, the consumption of cocaine and more precisely crack cocaine is a public health problem in Martinique [43]. The number of cases of severe poisoning was probably underestimated for several reasons, mainly the absence of systematic toxicological analysis and difficulties in the diagnosis. Indeed, in some cases, clinical signs were not specific and the history was poor.

Screening for the toxicant involved in severe poisoning is essential to guide specific therapeutics and to predict outcome. In our hospitals, the only available screenings are qualitative detection of tricyclic antidepressants, benzodiazepines, and barbiturates, and quantitative analysis of blood alcohol and acetaminophen concentrations. Pesticides/herbicides used in the West Indies for which screening tests are available (Cholinesterase test for Témik^®^, and dithionite test or blood concentration for paraquat) were not performed in our study. There were also no tests to investigate the presence of opiates, cannabis, and/or cocaine. Therefore, in our study, the diagnosis of acute poisonings was based on the history reported by the patient or their family and on the clinical manifestations and laboratory results. This underscores the underuse of the available toxicologic analysis in Martinique, which should be an integral part of the management.

Poisoning management includes symptomatic measures, organ support, and antidotes. In our study, 15% of patients received activated charcoal, 20% were treated with gastric lavage, and 21% received antidotes. The only treatment statistically associated with a favorable outcome was antidote administration. Indeed, the use of antidotes was systemic in pesticide/herbicide poisoning (atropine) and Rubigine^®^ intoxication (calcium gluconate) (Table 5). N-acetylcysteine and vitamin K were used in all cases of paracetamol poisoning. The use of flumazenil is not routine in benzodiazepine poisoning, because in most cases, benzodiazepine poisonings are associated with multiple other toxicants. This can be explained by a characteristic clinical picture and the awareness of the seriousness of these types of poisoning.

Our study had several limitations because of its retrospective design and its focus on severe patients only. However, this is the first epidemiological report from French West Indies. It confirms the burden of poisonings as a neglected public health problem and the need for better investigation and understanding of this topic. This study highlights the need to develop a toxicovigilance system in the West Indies with a toxicovigilance system for several reasons. The incidence of poisoning in the West Indies is unknown. Moreover, there is no regional or interregional registry to identify poisonings and allowing descriptive analyses. A local facility would permit the collection of epidemiological and clinical information and provide better management and follow-up of the severely poisoned patients. This system would allow monitoring of the toxic effects of products currently available in our region, setting up warning and prevention measures, informing and training the public and health professionals, and sensitizing regional and interregional authorities on specific toxicants like illegal drugs, chlordecone, Rubigine^®^, Gramoxone^®^, some caustics, pesticides, local fruits (like starfruit [13]), and some traditional plants (Table 5). The dissemination of this information could be made through teaching and training campaigns coordinated by a Caribbean toxicovigilance system.

## 6. Conclusions

Due to its tropical climate, its geographic location in the Caribbean, which allows the country to be supplied with substances prohibited under French law, but also due to non-compliance with regulations regarding the disposal of product stocks, Martinique has particular characteristics in terms of the epidemiology of severe acute poisoning. As evidenced in this study, mortality is high despite appropriate management. There is, therefore, an urgent need to provide French West Indies with an Anti-Poison Center and local toxicovigilance facility. Prospective multicenter studies in the region are required to better understand the epidemiology of poisonings in the region. Clinicians should also be aware that severe toxicity can result from exposure to non-authorized industrial drugs as well as to environmental agents including plants, herbal, and traditional remedies.

## Figures and Tables

**Figure 1 toxics-08-00028-f001:**
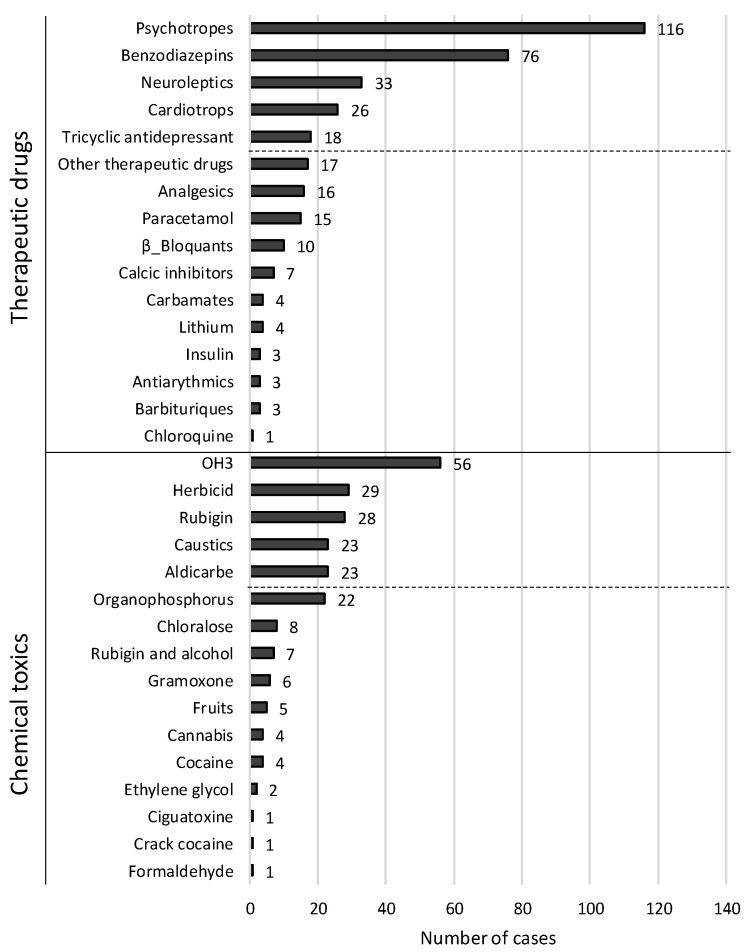
Frequency of the involved toxics.

**Figure 2 toxics-08-00028-f002:**
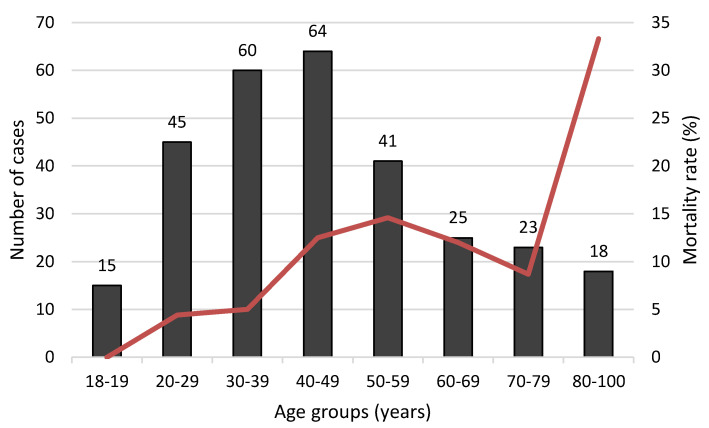
Number of cases and mortality rate according to age groups in our population.

**Figure 3 toxics-08-00028-f003:**
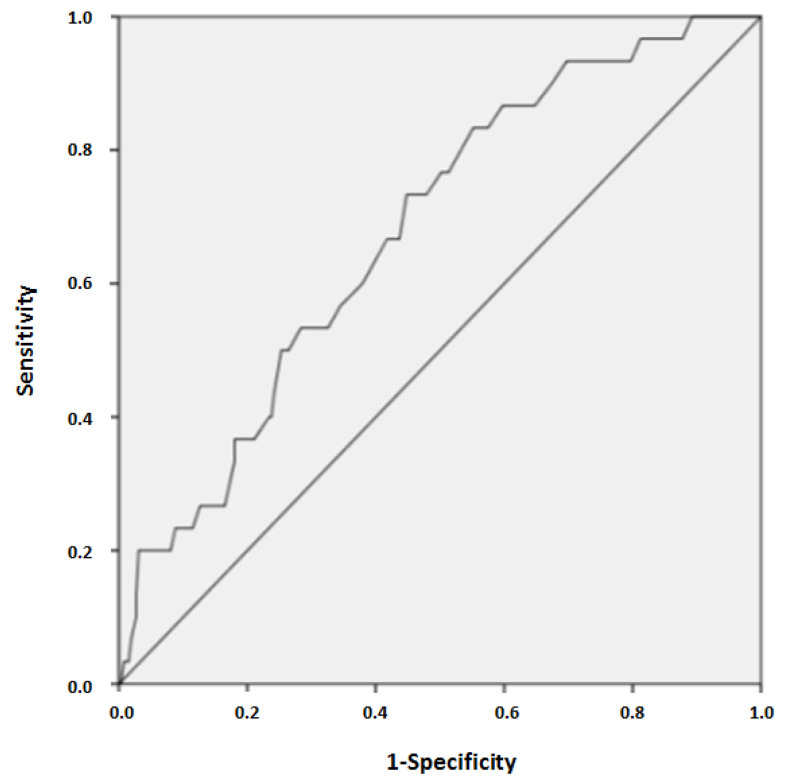
Receiver operating characteristic (ROC) curve showing the accuracy of age to predict mortality in severely poisoned patients (AUC: 0.678).

**Figure 4 toxics-08-00028-f004:**
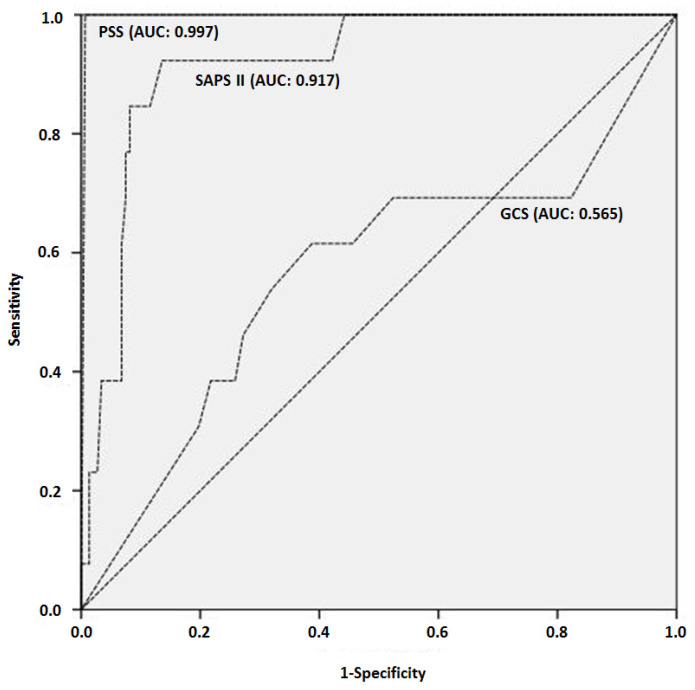
Receiver operating characteristic (ROC) curve showing the accuracy of Glasgow Coma Scale (GCS) at admission, Simplified Acute Physiology Score (SAPS) II score and Poisoning Severity Score (PSS)to predict mortality in severely poisoned patients.

**Table 1 toxics-08-00028-t001:** The epidemiological characteristics and outcome of patients.

Variable	Total Population		Non Survivors		Survivors		*p*
	Nb	Result	Nb	Result	Nb	Result	
Age (years)	291	46 ± 19	30	56 ± 18	261	45 ± 18	0.001
Male gender	291	166 (57%)	30	17 (56.7%)	261	149 (57.1%)	0.965
Medical history							
Psychiatric disorder	287	143 (49.8%)	30	14 (46.7%)	257	129 (50.2%)	0.865
Drug intoxication	284	65 (22.9%)	29	1 (3.4%)	255	64 (25.1%)	0.031
Alcohol abuse	284	59 (20.8%)	29	6 (20.7%)	255	53 (20.8%)	0.940
Hospitalization							
Emergency Department	287	185 (64.5%)	30	19 (63.3%)	257	166 (64.6%)	0.785
Medical ward	288	26 (9%)	29	4 (13.8%)	259	22 (8.5%)	0.271
Level 3 ICU	288	227 (78.8%)	30	23 (76.7%)	258	204 (79.1%)	0.802
Level 2 ICU	290	71 (24.5%)	30	6 (20%)	260	65 (25%)	0.787
Severity							
PSS	289	2.7 ± 0.8	30	4 ± 0	259	2.5 ± 0.7	0.000
SAPS II	191	39 ± 23	19	71 ± 19	172	36 ± 20	0.000
Issue							
Hospital Length of stay	290	6 ± 7	30	5 ± 8	260	6 ± 7	0.750
1 day	290	18 (6.2%)	30	8 (26.7%)	260	10 (3.8%)	-
2 to 7 days	290	225 (77.3%)	30	17 (56.7%)	260	208 (80%)	-
8 to 14 days	290	21 (7.2%)	30	3 (10%)	260	18 (6.9%)	-
>15 days	290	26 (8.9%)	30	2 (6.7%)	260	24 (9.2%)	-
Cardiac arrest	291	34 (11.7%)	30	30 (100%)	261	4 (1.5%)	0.000
Death	291	30 (10.3%)	30	30 (100%)	261	0 (0%)	-

-: *p* value not calculated.

**Table 2 toxics-08-00028-t002:** Involved toxics in survivors and non survivors.

Variable	Total Population		Non-Survivors		Survivors		*p*
	Nb	Result	Nb	Result	Nb	Result	
Therapeutic drug intoxication	291	169 (58.1%)	30	11 (36.7%)	261	158 (60.5%)	0.012
Nb of therapeutic drug involved	291	1.2 ± 1.3	30	0.6 ± 0.9	261	1.3 ± 1.3	0.090
Antipsychotics	291	116 (39.9%)	30	4 (13.3%)	261	112 (42.9%)	0.002
Benzodiazepines	291	76 (26.1%)	30	2 (6.7%)	261	74 (28.4%)	0.010
Barbiturates	291	3 (1%)	30	0 (0%)	261	3 (1.1%)	0.555
Neuroleptics	291	33 (11.3%)	30	1 (3.3%)	261	32 (12.3%)	0.144
Lithium	291	4 (1.4%)	30	0 (0%)	261	4 (1.5%)	0.495
Tricyclic antidepressants	291	18 (6.2%)	30	1 (3.3%)	261	17 (6.5%)	0.493
Carbamate	291	4 (1.4%)	30	0 (0%)	261	4 (1.5%)	0.495
Cardiotrops	291	26 (8.9%)	30	3 (10%)	261	23 (8.8%)	0.829
Antiarrhythmics	291	3 (1%)	30	0 (0%)	261	3 (1.1%)	0.555
Β-Blockers	291	10 (3.4%)	30	2 (6.7%)	261	8 (3.1%)	0.305
Calcium inhibitors	291	7 (2.4%)	30	1 (3.3%)	261	6 (2.3%)	0.726
Analgesics	291	16 (5.5%)	30	1 (3.3%)	261	15 (5.7%)	0.583
Paracetamol	291	15 (5.2%)	30	1 (3.3%)	261	14 (5.4%)	0.634
Chloroquine	291	1 (0.3%)	30	0 (0%)	261	1 (0.4%)	0.734
Insulin	291	3 (1%)	30	0 (0%)	261	3 (1.1%)	0.555
Other therapeutic drugs	291	17 (5.8%)	30	3 (10%)	261	14 (5.4%)	0.305
Chemical intoxication	291	152 (52.2%)	30	19 (63.3%)	261	133 (51%)	0.199
Nb of chemical products involved	291	0.8 ± 0.9	30	0.9 ± 0.9	261	0.7 ± 0.9	0.250
Herbicides	291	29 (10%)	30	6 (20%)	261	23 (8.8%)	0.053
Aldicarb	291	23 (7.9%)	30	0 (0%)	261	23 (8.8%)	0.090
Organophosphorus	291	22 (7.6%)	30	2 (6.7%)	261	20 (7.7%)	0.845
Gramoxone	291	6 (2.1%)	30	5 (16.7%)	261	1 (0.4%)	0.000
Chloralose	291	8 (2.7%)	30	0 (0%)	261	8 (3.1%)	0.331
Rubigine^®^	291	28 (9.6%)	30	4 (13.3%)	261	24 (9.2%)	0.467
Caustics	291	23 (7.9%)	30	1 (3.3%)	261	22 (8.4%)	0.327
Ethylene glycol	291	2 (0.7%)	30	0 (0%)	261	2 (0.8%)	0.630
Formaldehyde	291	1 (0.3%)	30	1 (3.3%)	261	0 (0%)	0.003
Crack cocaine	291	1 (0.3%)	30	0 (0%)	261	1 (0.4%)	0.734
Cocaine	291	4 (1.4%)	30	1 (3.3%)	261	3 (1.1%)	0.331
Cannabis	291	4 (1.4%)	30	0 (0%)	261	4 (1.5%)	0.495
OH3	291	56 (19.2%)	30	5 (16.7%)	261	51 (19.5%)	0.705
Fruits	291	5 (1.7%)	30	2 (6.7%)	261	3 (1.1%)	0.028
Ciguatoxine	291	1 (0.3%)	30	0 (0%)	261	1 (0.4%)	0.734
Mixed therapeutic and chemical toxic	291	30 (10.3%)	30	0 (0%)	261	30 (11.5%)	0.050
Nb of toxicant involved	291	1.1 ± 0.3	30	1 ± 0	261	1.1 ± 0.3	0.005
Multiple therapeutic drugs intoxication	169	61 (36.1%)	11	2 (18.2%)	158	59 (37.3%)	0.025
Multiple chemical intoxication	152	48 (31.6%)	19	7 (36.8%)	133	41 (30.8%)	0.371
Drug overdose	291	37 (12.7%)	30	7 (23.3%)	261	30 (11.5%)	0.065
Intentional self-poisoning	289	250 (86.5%)	30	24 (80%)	259	226 (87.3%)	0.484

**Table 3 toxics-08-00028-t003:** Clinical and biological parameters recorded in our patients.

Parameter	Normal Range	Total Population		Non Survivors		Survivors		*p*
		Number	Result	Number	Result	Number	Result	
Temperature (°C)	36.1–37.5	74	36.6 ± 1.6	11	36.2 ± 2.3	63	36.7 ± 1.4	0.361
Breath rhythm (breath/min)	16–20	50	25 ± 19	9	44 ± 37	41	21 ± 7	0.000
SaO_2_ (%)	92–100	94	94 ± 9	10	90 ± 10	84	94 ± 9	0.099
Cardiac rhythm (beat/min)	60–90	179	89 ± 28	23	85 ± 30	156	90 ± 28	0.472
Shock	-	291	78 (26.8%)	30	28 (93.3%)	261	50 (19.2%)	0.000
Ventricular fibrillation or tachycardia	-	291	8 (2.7%)	30	7 (23.3%)	261	1 (0.4%)	0.000
GCS	15	227	9 ± 4	21	9 ± 5	206	8 ± 4	0.860
Seizures	-	291	31 (10.7%)	30	5 (16.7%)	261	26 (10%)	0.260
Myoclonus	-	291	28 (9.6%)	30	1 (3.3%)	261	27 (10.3%)	0.217
Gastro-intestinal disorders	-	291	78 (26.8%)	30	14 (46.7%)	261	64 (24.5%)	0.010
Pneumonia	-	291	49 (16.8%)	30	7 (23.3%)	261	42 (16.1%)	0.315
Conjunctivitis	-	291	3 (1%)	30	0 (0%)	261	3 (1.1%)	0.555
Other infections	-	291	77 (26.5%)	30	10 (33.3%)	261	67 (25.7%)	0.173
Biology								
pH	7.40 ± 0.02	81	7.30 ± 0.20	17	7.20 ± 0.20	64	7.30 ± 0.20	0.092
Alkaline reserve (mmol/L)	22–29	66	19.3 ± 7.1	13	15.2 ± 6.3	53	20.3 ± 7	0.019
PaCO_2_ (mmHg)	35–45	68	39.8 ± 15	10	40.5 ± 18.8	58	39.6 ± 14.4	0.867
Lactate (mmol/L)	0.5–2.2	53	6.4 ± 5.9	9	11.1 ± 7.7	44	5.4 ± 5	0.006
Hyper lactacidemia	-	53	37 (69.8%)	9	8 (88.9%)	44	29 (65.9%)	0.171
Sodium (mmol/L)	136–145	59	139 ± 8	8	136 ± 9	51	139 ± 7	0.225
Potassium (mmol/L)	3.5–5.1	82	4 ± 1.3	12	5.4 ± 1.5	70	3.8 ± 1.1	0.000
Hypokalemia	-	82	26 (31.7%)	12	1 (8.3%)	70	25 (35.7%)	0.029
Hyperkalemia	-	82	13 (15.9%)	12	7 (58.3%)	70	6 (8.6%)	0.000
Calcium (mmol/L)	2.15–2.55	33	2.1 ± 0.4	2	1.4 ± 0.2	31	2.1 ± 0.4	0.029
Magnesium (mmol/L)	0.75–0.90	10	0.9 ± 0.2	2	0.8 ± 0.1	8	0.9 ± 0.3	0.497
Phosphorus (mmol/L)	0.80–1.35	22	1.1 ± 0.6	1	1.27	21	1.1 ± 0.6	0.805
Creatinine (µmol/L)	62–106	90	220 ± 213	15	270 ± 179	75	210 ± 219	0.317
Urea nitrogen (mmol/L)	1.7–8.3	73	15 ± 34	10	43 ± 87.8	63	10.6 ± 8.9	0.004
Creatine Kinase (IU/L)	38–174	34	15,974 ± 45,164	5	38,193 ± 44,310	29	12,143 ± 44,947	0.000
Rhabdomyolysis	-	34	20 (58.8%)	5	5 (100%)	29	15 (51.7%)	0.041
Troponins (µg/L)	0–0.014	15	0.8 ± 1.9	4	0.5 ± 0.6	11	0.9 ± 2.2	0.700
Elevated troponins	-	3	3 (100%)	1	1 (100%)	2	2 (100%)	0.187
Glycaemia (mmol/L)	3.9–7.1	12	8.3 ± 5.1	2	4.6 ± 6	10	9.1 ± 4.9	0.270
Aspartate aminotransferases (AST) (IU/L)	<37	43	1968 ± 3894	6	5452 ± 4317	37	1403 ± 3570	0.042
Alanine aminotransferases (ALT) (IU/L)	<40	43	961 ± 2211	6	2431 ± 2574	37	723 ± 2089	0.239
Platelet count (Giga/L)	150–400	29	152 ± 95	7	75 ± 74	22	177 ± 88	0.010
Prothrombine time (%)	60–100	38	42 ± 31	8	31 ± 18	30	45 ± 34	0.262
SvO_2_ (%)	75–85	2	82 ± 4.2	1	85	1	79	-
Toxicologic analysis	-	284	111 (39.1%)	30	8 (26.7%)	254	103 (40.6%)	0.218

-: Non appropriate.

**Table 4 toxics-08-00028-t004:** Therapeutic management of our patients.

Parameter	Total Population		Non Survivors		Survivors		*p*
	Nb	Result	Nb	Result	Nb	Result	
Activated charcoal	291	44 (15.1%)	30	2 (6.7%)	261	42 (16.1%)	0.172
Gastric lavage	291	57 (19.6%)	30	8 (26.7%)	261	49 (18.8%)	0.302
Antidotes	291	60 (20.6%)	30	2 (6.7%)	261	58 (22.2%)	0.046
Mechanical ventilation	291	171 (58.8%)	30	29 (96.7%)	261	142 (54.4%)	0.000
ARDS	172	12 (7%)	30	6 (20%)	142	6 (4.2%)	0.000
MV duration (days)	171	6 ± 8	29	5 ± 8	142	6 ± 8	0.433
Catecholamines	291	78 (26.8%)	30	28 (93.3%)	261	50 (19.2%)	0.000
Renal replacement therapy	291	32 (11%)	30	13 (43.3%)	261	19 (7.3%)	0.000
ECMO	291	3 (1%)	30	2 (6.7%)	261	1 (0.4%)	0.001
Treatment	291	133 (45.7%)	30	20 (66.7%)	261	113 (43.3%)	0.015

**Table 5 toxics-08-00028-t005:** Features of the main involved toxicants with a focus on those more rarely observed in Western countries.

**Main Toxicants Involved**	**Related Clinical and Biological Features**	
Benzodiazepines	Coma, aspiration pneumonia	
Neuroleptics	Coma, respiratory failure, shock, aspiration pneumonia, renal failure, rhabdomyolysis	
Cardiotrops	Cardiac arrhythmias, shock, heart conduction dysfunction, cardiac arrest, renal failure, elevation in lactate, rhabdomyolysis, liver failure	
Tricyclic antidepressants	Coma, seizures, shock, heart conduction dysfunction, aspiration pneumonia, renal failure, rhabdomyolysis	
**The Rarest Toxicants in Western Countries**	**Specific Features**	**Specific Management**
Herbicides, paraquat	Respiratory failure (acute respiratory distress syndrome), elevation in transaminase, renal failure	No effective antidote
Rubigine^®^ (Hydrofluoric acid)	Gastrointestinal disturbances (corrosion), dermal injuries, hypocalcemia, metabolic acidosis, respiratory failure, seizures, dysrhythmias, cardiovascular disorders, cardiac arrest	Calcium gluconate, analgesics, hemodialysis,
Pesticides Organophosphates	Excessive respiratory secretions, and bronchoconstriction, bradycardia, myosis, neuromuscular weakness, seizures, coma, cardiovascular collapse	Atropine, pralidoxime, diazepam
Star Fruit (Carambola) (Toxin: oxalic acid)	Hiccups, vomiting, encephalopathy, seizures, neuropsychiatric manifestations, acute kidney injury and chronic kidney disease Renal histology: oxalate crystals obstructing the tubules	No effective antidote Renal replacement therapy (hemodialysis or hemoperfusion)

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
