# Peer review of "Clinical and Epidemiological Characteristics of Severe Acute Adult Poisoning Cases in Martinique: Implicated Toxic Exposures and Their Outcomes"

_toxics, 2020, doi:10.3390/toxics8020028_

Round 1
Reviewer 1 Report
This is an interesting retrospective study which provides relevant information on the current characteristics of severe poisoning cases in Martinique. I have few comments that the authors may want to consider.
1- The table on the clinical symptoms and biological parameters does not really allow to identify which toxic was involved. Its clinical relevance is therefore quite limited. Would there be a simple way to give this information (at least for the most frequent toxic agents) in the same table without making it too difficult to read.
2- Same thing for the treatment. I realize that this is a lot of additional work, so I'll let the editor and the authors decide if it is worth doing.
3- It would have been interesting to isolate the toxic agents that are not or very rarely found in western Europe and to give their specificity and the treatment that was used (may be a separate table)
4- Minor comment: The text and the tables need to be edited very carefully and reviewed by a native English speaker.
Author Response
In our study, we aimed to give describe the global clinical and sociodemographic patterns of acute poisonings in Martinique. We highlight the urgent need to provide French West Indies with an Anti-Poison Center and local Toxicovigilance facility. We also emphasize the need for prospective multicentre studies in the region to better understand the epidemiology of poisonings. Finally, we alert clinicians about severe toxicities from exposure to non-authorized industrial drugs, and to environmental agents including plants, herbal and traditional remedies.
Dear Editor,
Please find enclosed the revised version of our article:
Clinical and epidemiological characteristics of severe acute adult poisoning cases in Martinique: implicated toxic exposures and their outcomes
You will find below an itemized list of our specific responses to the reviewers' comments with the corresponding changes made in the revision.
Reviewer 1 –
This is an interesting retrospective study which provides relevant information on the current characteristics of severe poisoning cases in Martinique. I have few comments that the authors may want to consider.
1- The table on the clinical symptoms and biological parameters does not really allow to identify which toxic was involved. Its clinical relevance is therefore quite limited. Would there be a simple way to give this information (at least for the most frequent toxic agents) in the same table without making it too difficult to read.
Response: We preferred to maintain all the details to allow the reader get all information he is searching. However, to facilitate the understanding, we added a table as suggested with the features that could be attributed to the most frequent toxicants.
2- Same thing for the treatment. I realize that this is a lot of additional work, so I'll let the editor and the authors decide if it is worth doing.
Response: We thank the reviewer for his comments. We preferred to maintain all the details to allow the reader get all information he is searching.
3- It would have been interesting to isolate the toxic agents that are not or very rarely found in western Europe and to give their specificity and the treatment that was used (may be a separate table)
Response: This was presented in a table as requested.
4- Minor comment: The text and the tables need to be edited very carefully and reviewed by a native English speaker.
Response: A native English speaker, as requested edited the manuscript.
Table – Features of the main involved toxicants with a focus on those more rarely observed in western countries
Main toxicants involved |
Related clinical and biological features |
|
Benzodiazepines |
Coma, aspiration pneumonia |
|
Neuroleptics |
Coma, respiratory failure, shock, aspiration pneumonia, renal failure, rhabdomyolysis |
|
Cardiotrops |
Cardiac arrhythmias, shock, heart conduction dysfunction, cardiac arrest, renal failure, elevation in lactate, rhabdomyolysis, liver failure |
|
Tricyclic antidepressants |
Coma, seizures, shock, heart conduction dysfunction, aspiration pneumonia, renal failure, rhabdomyolysis |
|
The rarest toxicants in western countries |
Specific features |
Specific management |
Herbicides, paraquat |
Respiratory failure (acute respiratory distress syndrome), elevation in transaminase, renal failure |
No effective antidote |
Rubigine® (fluorhydric acid) |
Gastrointestinal disturbances (corrosion), dermal injuries, hypocalcemia, metabolic acidosis, respiratory failure, seizures, dysrhythmias, cardiovascular disorders, cardiac arrest |
Calcium Gluconate, analgesics, hemodialysis,
|
Pesticides Organophosphates
|
Excessive respiratory secretions, and bronchoconstriction, bradycardia, myosis, neuromuscular weakness, seizures, coma, cardiovascular collapse |
Atropine, pralidoxime, diazepam |
Star Fruit (Carambola) (Toxin : oxalic acid)
|
Hiccups, vomiting, encephalopathy, seizures, neuropsychiatric manifestations, acute kidney injury and chronic kidney disease Renal histology: oxalate crystals obstructing the tubules |
No effective antidote Renal replacement therapy (hemodialysis or hemoperfusion) |

Reviewer 2 Report
This manuscript provides a 10 year epidemiologic overview of adult poisonings in the French west indies. The paper is well written. The Intro and discussion is well research and referenced. My only concern is that the data is already almost 10 years old. I do think that the content is significant for public health leaders for this country to understand the need for some type of service that provides recommendations to citizens with poisonings and toxic exposures as well as to highlight that a problem does exist.
Author Response
Reviewer 2 -
This manuscript provides a 10 year epidemiologic overview of adult poisonings in the French west indies. The paper is well written. The Intro and discussion is well research and referenced. My only concern is that the data is already almost 10 years old. I do think that the content is significant for public health leaders for this country to understand the need for some type of service that provides recommendations to citizens with poisonings and toxic exposures as well as to highlight that a problem does exist.
Response:
We thank the reviewer for his encouraging comments. We agree that the data provided in this study is already 10 years old. However, as explained, our data may help our public health leaders to better understand the needs to prevent and manage poisonings and toxic exposures in our island. There is a need for a large prospective study.